# Uncovering the Role of Epstein–Barr Virus Infection Markers for Remission in Rheumatoid Arthritis

**DOI:** 10.3390/biomedicines11092375

**Published:** 2023-08-24

**Authors:** Ana Banko, Andja Cirkovic, Ivica Jeremic, Milica Basaric, Milka Grk, Rada Miskovic, Ivana Lazarevic, Danijela Miljanovic

**Affiliations:** 1Institute of Microbiology and Immunology, Faculty of Medicine, University of Belgrade, 11000 Belgrade, Serbia; ivana.lazarevic@med.bg.ac.rs (I.L.); danijela.karalic@med.bg.ac.rs (D.M.); 2Institute for Medical Statistics and Informatics, Faculty of Medicine, University of Belgrade, 11000 Belgrade, Serbia; andja.cirkovic@med.bg.ac.rs; 3Faculty of Medicine, University of Belgrade, 11000 Belgrade, Serbia; ivicaje@yahoo.com (I.J.); rada_delic@hotmail.com (R.M.); 4Institute of Rheumatology, Faculty of Medicine, University of Belgrade, 11000 Belgrade, Serbia; basaric.milica@gmail.com; 5Institute of Human Genetics, Faculty of Medicine, University of Belgrade, 11000 Belgrade, Serbia; milkagrk@gmail.com; 6Clinic of Allergy and Immunology, University Clinical Center of Serbia, 11000 Belgrade, Serbia

**Keywords:** Epstein–Barr virus (EBV), rheumatoid arthritis (RA), marker, anti-EBNA-1 IgG, anti-EBV antibodies, methotrexate (MTX), tumor necrosis factor (TNF)-alpha inhibitor

## Abstract

Epstein–Barr virus (EBV) infection has been shown as a potential risk factor for the development of rheumatoid arthritis (RA). This prospective research aimed to investigate whether EBV infection markers changed during the six-month follow-up period in 133 RA patients (80 newly diagnosed on methotrexate (MTX)—RA-A, and 53 on biologic therapy—RA-B) and whether it was related to a disease outcome. Reduction of disease activity and inflammation was obtained. A significant decline in seroprevalence and titer for anti-VCA-IgM (*p* = 0.022 and *p* = 0.026) and anti-EA(D)-IgM (*p* = 0.022 and *p* = 0.006) in RA-A, and in seroprevalence and titer of anti-EA(D)-IgG in the RA-B subgroup (*p* = 0.021 and *p* = 0.006) were detected after the follow-up. A lower titer of anti-EBNA1-IgG could be considered a significant marker of RA remission in all RA patients regardless of age and gender (OR = 0.99, 95% CI OR = 0.98–0.99, *p* = 0.038), and also in RA-B patients separately (OR = 0.988, 95% CI OR = 0.98–0.99, *p* = 0.041). This study supported the basic hypothesis that the immune response to EBV infection is involved in the RA pathogenesis, at the beginning of the disease or during the RA evolution. Moreover, the potential influence of MTX or TNF-alpha inhibitors on the impairment of the host to control EBV infection was indirectly refuted.

## 1. Introduction

Rheumatoid arthritis (RA) is an autoimmune disease characterized by chronic systemic inflammation and the involvement of the synovial joints. The prevalence is 0.5–1% in the general population, with a female-to-male ratio of 2:1 to 3:1, and the incidence peaks between 40 and 60 years [1]. Despite RA’s annual incidence of 25–50 cases per 100,000 population and its significant impact on the quality of life, this disease still has an insufficiently clarified etiological background [2]. Similarly to other autoimmune disorders, it seems that the interplay between various genetic, immune, hormonal and environmental risk factors contributes to the onset and pathogenesis of RA [2]. Among the environmental factors, infections are of particular importance. Thus, Epstein–Barr virus (EBV) infection, with multiple mechanisms in driving autoimmunity, has been shown as a potential trigger for the development of inflammatory arthritis, too [3].

EBV is a member of the Human Herpesviruses family. Known as one of the most prevalent human viruses, it latently infects up to 99% of the population worldwide [4]. Superior viral capacities for lifelong survival in human cells represent a unique challenge for the immune system [5]. After the primary lytic infection of epithelial cells of the oropharyngeal cavity, the EBV lifecycle includes latency in B lymphocytes with occasional reactivations. This kind of existence is possible due to the controlled expression of viral genes, from the very limited number in latency to the total expression during lytic reactivations [6]. Moreover, the latency itself is classified into separate categories associated with different diseases, depending on the number of expressed genes. In response to primary contact with the virus, the following antiviral antibodies are synthesized: to viral capsid antigen (CA), early antigen (EA) and EBV nuclear antigen 1 (EBNA1) EA IgG levels persist up to 2 years following primary infection, with possible detectability during reactivations [7]. CA IgG and EBNA1 IgG persist for life, with the levels of CA IgG higher during lytic infection (primary or reactivation) [7].

The involvement of EBV infection in RA has long been suggested, mostly according to virus association with several other autoimmune diseases: multiple sclerosis, systemic lupus erythematosus, Sjögren’s syndrome, autoimmune thyroiditis, inflammatory bowel diseases, etc. [8]. The underlying mechanism hypothesis is based on the imbalance between EBV infection and the host’s capacity to control it and molecular mimicry between viral antigens and self-antigens. Cross-reactivation between circulating antibodies to viral antigens and self-antigens and a decreased T cell response to EBV regulatory protein gp110 (BALF4) in RA patients lead to impairment to limit viral replication, chronic expression of EBV antigens and sustained inflammatory response [9,10,11]. Finally, previous researchers demonstrated not only the EBV presence in synovial cells of chronic RA but also higher rates of anti-citrullinated protein antigen (ACPA) producing B cells in synovium when active infection is present [9]. In addition, it was also shown that rheumatoid factor (RF) could reactivate EBV in RA synovium [12].

Despite the theories and indications, there is no clear confirmation that previous infection predisposes to RA [2,13]. However, it is still of primary importance that higher frequencies and levels of anti-EBV antibodies in RA have been proven many times [5,14,15]. Moreover, in RA patients, EBV DNA was detected more frequently in PBMCs, synovial fluid or saliva than in patients with non-RA inflammatory diseases or healthy controls [16]. Described EBV activity could be of significance for a greater susceptibility to developing lymphomas, which has been proven to be more frequent in RA patients [17]. The influence of RA treatment with methotrexate (MTX) or TNF blockers and other biological treatments on the risk of developing EBV-associated complications, such as lymphomas, is still debatable. It was demonstrated that therapy protocols that included MTX could increase EBV loads, but not when they included TNF blockers [18]. However, long-term treatment with MTX did not influence viral load, Moreover, new biological treatments such as abatacept and tocilizumab did not significantly modify EBV loads, or even reduced them [19]. Still, this does not exclude a reduced risk of lymphoma.

To improve the understanding of EBV infection dynamics during the course of RA, the goals of this research were to investigate the status of EBV infection and whether its changes during the six-month follow-up period are influenced by the different therapy approaches to RA patients and to evaluate whether some of the markers of EBV infection could be considered as markers of RA development or remission.

## 2. Materials and Methods

### 2.1. Study Design and Participants

This prospective cohort study included 133 RA patients diagnosed and treated at the Institute of Rheumatology, University Clinical Centre in Belgrade, between June 2020 and November 2022. RA diagnosis was established according to the American College of Rheumatology (ACR) and EULAR criteria [20]. Patients under 18 years of age, unable to cooperate, or with significant comorbidities (severe cardiac, pulmonary, and psychiatric diseases) or malignancies were excluded from the study. This study successively recruited all available RA patients who met the predefined inclusion criteria.

All included RA patients were subdivided into two groups. The first subgroup consisted of newly diagnosed RA patients (RA-A, *n* = 80). They were treated with methotrexate during the 6-Month follow-up. Low doses of systemic glucocorticoids (≤10 mg), paracetamol, and non-steroid anti-inflammatory drugs (NSAIDs) were administered occasionally. Another subgroup included RA patients with inadequate response to the first-line therapy (RA-B, *n* = 53). These patients were treated with methotrexate for at least 6 months and met criteria for starting biological drugs according to local Serbian regulations (disease activity index DAS28 > 5.1). All patients received tumor necrosis factor (TNF)-alpha inhibitor in combination with methotrexate. Low-dose systemic steroids and NSAIDs were allowed.

After detailed clinical interviews and physical examination of patients, all relevant demographic, clinical, laboratory, and virology data were collected at baseline and 6 months later as well. Loss to follow-up was 25% (33 out of 133).

Primary endpoint was remission defined as DAS28-CRP < 2.6 while secondary endpoint was remission defined according to at least one of the following criteria: DAS28-ESR, DAS28-CRP, SDAI, or CDAI. It was considered for DAS28-ESR < 2.6, DAS28-CRP < 2.6, SDAI ≤ 3.3, or CDAI ≤ 2.8. 

All participants provided written informed consent. The study was performed in accordance with the Declaration of Helsinki and was approved by the Ethical Board of the Faculty of Medicine, University of Belgrade (No 1550/IX-14) and by the Ethical Board of the Institute of Rheumatology, Belgrade (No 29/1-31).

### 2.2. Samples

RA patients’ serum and whole blood samples were collected at the Institute of Rheumatology, Belgrade. Control group serum and whole blood samples were obtained from healthy volunteers. After collecting 5 mL of blood in plain vacutainers, sera were separated by centrifugation. Plasma was also separated by centrifugation from 5 mL of ethylenediaminetetraacetic acid (EDTA) blood collected in vacutainer tubes. Two tubes (sera and plasma) from each patient were immediately tested or stored at −70 °C until further analysis.

### 2.3. EBV Serological Testing

Anti-EBV antibodies against CA (IgG), CA (IgM), EA (IgG), EA (IgM), and EBNA1 (IgG) were detected and measured in collected sera using commercial ELISAs according to the manufacturer’s instructions (Euroimmun, Lubeck, Germany). Standard calibrators were used in each assay to calculate index values/optical density (OD) ratios, which served as a quantitative measure of IgG antibody levels or a semi-quantitative measure of IgM antibody levels. All assays met pre-determined quality control measures based on positive, negative, and blank controls. The positivity of IgG antibody presence was defined by a cut-off value of 20 relative units (RU/mL). The positivity of IgM antibody presence was defined as OD ratio ≥ 1.1. Absorbances were recorded on a Multiscan FC microplate reader (Thermo Scientific, Waltham, MA, USA) using a wavelength of 405 nm with background subtraction at 650 nm.

### 2.4. EBV DNA Detection

Viral DNA was isolated from 200 μL plasma using a PureLink Genomic DNA Mini Kit (Invitrogen by Thermo Fisher Scientific, Waltham, MA, USA) according to the manufacturer’s instructions. Two hundred thirty-three DNA isolates were further used in a nested-PCR method to amplify the C terminus of the EBNA1 EBV gene, as described previously [21,22].

### 2.5. Data Analysis

Descriptive statistics were used in order to present the data. Arithmetic mean with standard deviation or median with range (from minimum to maximum values) were applied for numerical data, depending on the distribution. Normal distribution was evaluated by Shapiro–Wilk test and box-plot. Absolute and relative numbers in percentages were in use for categorical data.

Student’s *t*-test for two independent samples or Mann–Whitney *U* test was used to compare numerical data between study subgroups, depending on the data distribution. The chi-square test was used to test the difference in the distribution of categories in two independent samples. McNemar’s test was applied to compare dichotomous data in the dependent sample. The Wilcoxon signed rank test was used to assess differences in numerical data without normal distribution at baseline and after 6 months of follow-up.

To evaluate predictors of remission in RA patients and RA-A and RA-B subgroups at 6-Month follow-up, first univariate, then multivariate logistic regression analyses were performed, adjusting for age and gender, reporting the risk ratio (RR), 95% confidence interval of the risk ratio (95% CI RR), and *p* value.

The whole analysis was performed in statistical software IBM Corp. Released 2019. IBM SPSS Statistics for Windows, Version 26.0. Armonk, NY: IBM Corp. All statistical methods considered significant for the chosen level of confidence of 0.05.

As this study is a part of a larger project, the sample size for the actual aim was not calculated. Instead, the power of the study was estimated by G Power 3.1.9.2 using the data obtained for the primary outcome (mean value of anti-EBNA1 IgG antibody titer in RA patients who did and who did not enter remission after 6-Month follow-up). For two-sided way of conclusion within Mann–Whitney test, error of the first type α = 0.05, sample size of 51 per group, and calculated Cohen’s d effect size 0.5, the estimated power of the study was 70.6%.

## 3. Results

### 3.1. Socio-Demographic Characteristics of RA Patients

A total of 133 seropositive RA patients with an average age of 58.86 ± 11.78 years and 2.5 times more females than males were included in this study. Almost 60% of all patients had secondary education. The distribution of smokers and non-smokers was equal. Eighty were newly diagnosed RA patients (RA-A), and the remaining 53 were RA patients with inadequate disease control who had met criteria for biological therapy (RA-B). There was no difference in evaluated socio-demographic characteristics between RA-A and RA-B patients except in smoking habit. There were significantly more smokers among RA-A than among RA-B patients (64% vs. 36%, *p* = 0.002). All detailed characteristics are presented in Table 1. Forty percent of all RA patients had a positive family history of this disease. Common non-rheumatic comorbidities were hypertension (46%), cardiovascular events (28%), and diabetes mellitus (13.5%).

### 3.2. EBV Infection Status during the 6 Month Follow-Up

In order to evaluate EBV infection status in RA patients at admission and 6 months after, serological and PCR DNA analyses were performed.

Evaluating EBV serological data of all RA patients during the defined time interval, we found a significant reduction in seroprevalence for anti-VCA IgM and anti-EA(D) IgM after 6 months (*p* = 0.004 and *p* = 0.031, respectively) (Table 2). Out of 17 anti-VCA IgM positive patients at baseline, only one remained positive, and 16 became negative. As well, out of 17 patients who were positive for anti-EA(D) IgM at baseline, three were still positive, whereas 14 became negative after 6-Month follow-up. Analyzing the two groups of patients separately (RA-A and RA-B), seroprevalence of anti-VCA IgM and anti-EA(D) IgM showed a significant reduction in RA-A patients (*p* = 0.022 and *p* = 0.022, respectively), while a significant decrease in anti-EA(D) IgG seroprevalence was detected in the RA-B subgroup (*p* = 0.021) during the 6-Month follow-up.

After evaluating the titer change in anti-EBV antibodies during the 6-Month follow-up, we obtained a significant decrease in anti-VCA IgM, anti-EA(D) IgG, and anti-EA(D) IgM (*p* = 0.006, *p* = 0.027, and *p* = 0.006, respectively) in all RA patients. RA-A patients showed a significant decrease in anti-VCA IgM and anti-EA(D) IgM (*p* = 0.026 and *p* = 0.006, respectively), while in RA-B patients, the titers for anti-VCA IgG and anti-EA(D) IgG decreased significantly during the 6-Month follow-up (*p* = 0.006 and *p* = 0.006, respectively). Data are shown in Table 2.

Four EBV EBNA1 DNA positive RA patients at baseline (4%) became negative after 6-Month follow-up, but this change remained without statistical significance (*p* = 0.687). In addition, two patients became EBV DNA positive after 6 months, although their results were negative at baseline. Moreover, there was no difference in EBV EBNA1 DNA presence during the 6-Month follow up in RA-A and RA-B subgroups (*p* = 0.087 and *p* = 0.375).

To determine the status of EBV infection, the overall results of both molecular and serological tests were analyzed. The active EBV infection was declared if EBV DNA or anti-EBV IgM antibodies were present. Active EBV infection was detected in 40% of patients at baseline, 20% of them transitioning to latent EBV infection after 6 months with no statistically significant change (*p* = 1.000). There was no difference in the status of EBV infection in either the RA-A or RA-B subgroups (*p* = 1.000 and *p* = 0.087, respectively).

### 3.3. Clinical, Laboratory, and Therapeutical Characteristics and Their Changes during the 6 Month Follow-Up

We observed an overall reduction in evaluated parameters relating to disease activity and inflammation during the 6-Month follow-up. Also, there was a significant change in RAID, RAQoL, and HAQ life indices showing an improvement in quality of life after the follow-up period (Table 3). The same results were obtained for RA-A and RA-B subgroups.

A total of 13 (10%) RA patients were in remission at the baseline according to DAS28-ESR (*n* = 1), DAS28-CRP (*n* = 4), SDAI (*n* = 8) or CDAI (*n* = 4) criteria, of which 10 met one, two met two, and only one met three criteria. Otherwise, 51 (50%) were in remission after 6-Month follow-up according to DAS28-ESR (*n* = 33), DAS28-CRP (*n* = 43), SDAI (*n* = 39) or CDAI (*n* = 31) criteria, of which five met one, 16 met two, 11 met three, and 19 met four criteria. There was a significant increase in the number of RA patients in remission after the treatment (*p* < 0.001). The conclusion was the same in both RA-A and RA-B patients (*p* = 0.001 and *p* < 0.001, respectively).

### 3.4. Prediction of RA Remission

The titer of anti-EBNA1 IgG was significantly lower at baseline in RA patients who entered than in those who did not enter remission after 6 month follow-up (144 (0–200) in remission vs. 173 (0–200) without remission, *p* = 0.015).

This analysis was followed by determination of a weak negative association between anti-EBNA1 IgG titer at baseline and CDAI score, as the measure of RA disease activity, after 6-Month follow-up (*ρ* = −0.171, *p* = 0.049). That suggested a possible relationship between anti-EBNA1 IgG antibodies at baseline and achievement of RA remission.

Finally, we performed a logistic regression analysis in order to analyze factors associated with RA remission. RA remission was defined according to DAS28-CRP as the primary endpoint. Also, RA remission was defined as the presence of at least one of the following four criteria: DAS28-ESR, DAS28-CRP, SDAI, and CDAI as secondary endpoint. In all RA patients, anti-EBNA1 IgG antibody titer levels at baseline could be considered as a potential marker regardless of age and gender (OR = 0.998, 95% CI OR = 0.98–0.99, *p* = 0.030). Also, the same anti-EBV antibody titer level could be considered as a marker of secondary endpoint regardless of age and gender (OR = 0.99, 95% CI OR = 0.98–0.99, *p* = 0.038). This parameter was not a significant predictor of remission in newly diagnosed RA patients, but it could be considered as a significant marker of the primary endpoint as well as secondary endpoint in RA patients on biological therapy regardless of age and gender (OR = 0.986, 95% CI OR = 0.97–0.99, *p* = 0,016 and OR = 0.988, 95% CI OR = 0.98–0.99, *p* = 0.041).

## 4. Discussion

Therapeutic options for the treatment of rheumatoid arthritis have evolved significantly over the past 20 years. However, considering the increased activity of EBV in RA patients, it was suspected that widely used methotrexate, with later introduced TNF alpha inhibitors, could also increase the risk of developing lymphoma [23]. Therefore, several papers, mostly published in the last decade, were dedicated to monitoring EBV activity during the treatment period, between 3 months and 5 years [18,19,23,24,25]. Despite the clearly defined hypothesis, the evidence obtained so far mostly refutes it [16]. On the other hand, there is an increased risk of developing lymphoproliferative complications in RA patients, including malignant lymphoma [26]. Considering that they are mostly related to the impaired control of chronic EBV infection, the need for missing knowledge arises [27].

It is not difficult to notice that previously mentioned studies that monitored EBV activity during the treatment of RA using only molecular markers of infection, including viral load, were investigated [18,19,23,24,25]. Observation of changes in the seroprevalence or level of anti-EBV antibodies was usually not performed. Thus, our study is rare in that it evaluated the dynamics of seropositivity and changes in anti-EBV antibody titer during a defined therapeutic follow-up of RA patients. The significant decrease in serological markers of active EBV infection during the treatment period, and in particular, the identification of a lower titer of anti-EBNA1 IgG as a marker of entering into RA remission, provides a new perspective on the virus–host relationship.

Although our study prospectively followed not the onset of RA but rather the dynamics of serological markers of EBV infection during the RA course, the obtained results indirectly suggested involvement of viral infection in the development of the disease. Both anti-VCA IgM and anti-EA(D) IgM are markers of active EBV infection. Thus, if only RA-A (the group of newly diagnosed patients) was observed, a significant reduction in seroprevalence and titer levels of anti-VCA IgM and anti-EA(D) IgM after 6 months from RA diagnosis might indicate an important role of the immune response to viral infection in the very beginning of the disease. The revealed results provide additional evidence for recently published research that explored EBV activity in the preclinical period of RA [28]. The authors showed elevated preclinical anti-EA(D) IgG levels in RA compared to controls. Moreover, increased anti-EA(D) IgG titers significantly correlated with increasing RF-IgM levels in future RA cases but not in controls. As the elevation of anti-EA(D) IgG titers could represent the consequence of viral activity, persistent or reactivated [29], previous researchers assumed that EBV reactivation is associated with the future development of RA [28]. Guided by their suggestions, and interpreting the significant decrease in anti-EA(D) IgG titers in the RA-B population from our study, it also could be supposed that RA patients with inadequate response to first-line therapy (which included methotrexate for at least 6 months) and disease activity index-DAS28 > 5.1 at the same time failed to control reactivated or persistently active EBV infection. However, in the sequence of events that did not lead to remission, it is unclear which of these two things would be the cause and which the consequence. In addition, it could be pointed out that the inclusion of tumor necrosis factor (TNF)-alpha inhibitor in the therapy protocol for these patients resolved RA disease activity simultaneously with the decline in serologic indicators of active EBV infection. Finally, EA(D) IgG can also be found in 85% of primary infections and 20–30% of past infections, so simultaneous analysis with other serological parameters is necessary [29].

Achieving RA remission in our study was followed by significant changes in findings for anti-EBV antibodies. In particular, there was a significant reduction in prevalence and titer levels of anti-VCA IgM and anti-EA(D) IgM antibodies during 6-Month follow-up. This result is important because it supports previous theories that immunosuppressive drugs like methotrexate do not necessarily stimulate viral replication [16,30,31]. On the other hand, there is evidence that lymphoproliferative disorders develop during the suppression of immune surveillance by MTX [27]. Further, when the effect of the TNF-alpha inhibitor was analyzed, RA remission was also followed by declining anti-EA(D) IgG, which could indicate resolution of persistent or reactivated EBV infection. Thus, it might be suggested that the use of TNF-alpha inhibitor did not lead to the inability to control EBV infection or at least to the inhibition of the immune response, either. Our results were also aligned with data obtained in a large cohort study that monitored EBV load, but not anti-EBV antibodies, and some other published reports [24,31] (Miceli-Richard 2009, Westergaard 2015).

Some of the first serological findings about the relationship between EBV infection and RA were reported in 1978, demonstrating that anti-EBNA-1 antibodies differ between RA patients and healthy controls [32]. Moreover, this discrimination was later proven in relation to other systemic autoimmune diseases like systemic lupus erythematosus (SLE) [15]. One explanation for the altered immune response to viral infection and the development of autoantibodies is based on the reported association between anti-EBNA-1 antibodies and the presence of rheumatoid factors and citrullinated peptides [31,33]. Despite those reports, two meta-analyses did not yield a significant association between anti-EBNA-1 antibodies and RA compared to the control group. However, one showed a significant association between RA and EBV, in particular, anti-VCA IgG and anti-EA IgG antibodies [2,13]. One of the latest studies also revealed that EBNA-1 antibody levels differed between RA patients, and also their relatives strongly predisposed to RA, and healthy subjects [14]. The authors recommended that the elevation of anti-EBNA-1 might serve as an early serological risk marker for future RA development if further prospective studies reach similar results. Unfortunately, three other prospective studies on preclinical RA elevation of anti-EBNA-1 IgG did not show any association [28,34,35]. All of the conflicting results, together with the absence of prospective studies, support the need for further investigation of this connection. Our study did not show a decrease in the level of EBNA-1 antibodies during the follow-up. When added to the previous obscure knowledge, it might be more indicative that those antibodies could not serve as a risk marker for RA onset. Moreover, there was no evidence of an association between anti-EBNA-1 antibody levels and immunosuppressive treatment including MTX, not only in our study but also in previous studies [31].

Svendsen et al. assumed that aberrant immune responses in the RA course, especially in more imminent RA when assessed prospectively, reduce anti-EBNA-1 antibody production [14]. Our study for the first time showed some additional points. The lower titer of anti-EBNA1 IgG antibodies was shown as a significant marker of RA remission regardless of age and gender. This marker was significant when all studied RA patients were considered, and when those on biological therapy were separated as a single group. Considering that low anti-viral antibody levels could reflect previous low viral replication and low viral load, these results might give a different context to Svendsens’ suggestion [36].

This study did not prove EBV viral load levels, as some of the studies before demonstrated even 10-fold higher EBV loads in RA patients than in healthy controls [37]. At the same time, this result was not an exception among available data in the literature [38]. The discrepancy in findings could be addressed by the selection of sample type for viral DNA testing. Thus, the majority of previous reports referred to peripheral B cells, PBMC and even synovial fluid or saliva [16,39,40]. On the other hand, our study detected EBV DNA from blood compartments free of cells, depicting the viral elements released during active replication and tendentially eliminating the episomes resting in latently infected B cells. If there was an intention to include viremia in markers of RA activity or markers for the development of lymphoproliferative complications, then there would be a still undefined standardization of sample selection and viremia levels that are clinically relevant. This delicate question is broader than RA itself and includes other immunocompromising conditions as well [41].

Some limitations should be considered. Inclusion of the parallel testing on synovial fluid might also be relevant [14]. Detection of EBV DNA by more sensitive and newer methods such as ddPCR could, according to some of the latest data in the literature, give new insight into low viremia values [42]. Loss to follow-up should be smaller. Unfortunately, this study took place at the same time as the COVID-19 pandemic, which was the reason for this loss to follow-up. For more comprehensive results and more precise conclusions, longer follow-up is necessary. However, the strength of this study is that it is among the first with this type of design, and following the obtained results after 6 months of follow-up, it would be useful to repeat the follow-up after a longer predefined period.

## 5. Conclusions

This study supported the basic hypothesis that EBV infection is involved in RA pathogenesis, but it is still unclear whether that is directly through viral activity or indirectly through an aberrant or modified immune response to viral infection. A significant reduction in seroprevalence and titer levels of anti-VCA IgM and anti-EA(D) IgM after 6 months from RA diagnosis might indicate an important role of the immune response to viral infection at the beginning of the disease. In addition, a significant decrease in anti-EA(D) IgG titers in the RA-B population from our study could also suggest that inadequate response to methotrexate with disease activity index-DAS28 > 5.1 is followed by failed control of reactivation and/or persistent EBV infection. Finally, the first demonstration of RA regression predictors among anti-EBV antibodies shines new light on the still unclear etiology mechanism. Thus, a lower titer of anti-EBNA1 IgG antibodies was shown as a significant marker of RA remission, as the primary and secondary endpoint when all studied RA patients were considered, especially those on biological therapy regardless of age and gender. For the potential inclusion of this finding in the diagnostic RA markers, longer and larger prospective studies are needed.

## Figures and Tables

**Table 1 biomedicines-11-02375-t001:** Socio-demographic characteristics of RA patients.

Characteristic	Total	Subgroups		*p* *
	*n* = 133	RA-A *n* = 80	RA-B *n* = 53	
Age (years), mean ± sd	58.86 ± 11.78	59.45 ± 12.73	57.96 ± 10.21	0.478 §
Gender, *n* (%)				0.769 €
Male	37 (27.8)	23 (28.8)	14 (26.4)	
Female	96 (72.2)	57 (71.3)	39 (73.6)	
BMI, mean ± sd	25.21 ± 4.32	25.34 ± 4.34	25.03 ± 4.34	0.684 §
Educational level, *n* (%)				0.058 €
Primary	25 (18.8)	14 (17.5)	11 (20.8)	
Secondary	78 (58.6)	53 (66.3)	25 (47.2)	
Tertiary or higher	30 (22.6)	13 (16.3)	17 (32.1)	
Smoking status, *n* (%)				0.002 €
Smoker	70 (52.6)	51 (63.8)	19 (35.8)	
Non-smoker	63 (47.4)	29 (36.3)	34 (64.2)	
Smoking duration (years), med (min-max)	30 (1–60)	30 (2–60)	30 (1–53)	0.313 ¥

***** for the level of significance of 0.05 according to Student *t*-test §, Mann–Whitney test ¥ or Chi-square test €.

**Table 2 biomedicines-11-02375-t002:** Seroprevalence of anti-EBV antibodies during the 6-Month follow-up.

	RA					
Anti-EBV Abs	Seroprevalence			Titer		
Time	Baseline	6 Month Follow-Up	*p* *	Baseline	6 Month Follow-Up	*p* *
**anti-EBNA-1 IgG**	95 (95.0)	96 (96.0)	1.000	149 (0–200)	161 (0–200)	0.310
**anti-VCA IgG**	99 (99.0)	99 (99.0)	1.000	184 (0–200)	189 (0–200)	0.083
**anti-VCA IgM**	17 (17.0)	1 (1.0)	0.004	0 (0–5.57)	0 (0–2.55)	0.006
**anti-EA(D) IgG**	21 (21.0)	31 (31.0)	0.076	0 (0–200)	0 (0–200)	0.027
**anti-EA(D) IgM**	17 (17.0)	7 (7.0)	0.031	0 (0–7.30)	0 (0–3.51)	0.006
	**RA-A**					
**anti-EBV Abs**	**Seroprevalence**			**Titer**		
**Time**	**Baseline**	**6 Month Follow-up**	***p* ***	**Baseline**	**6 Month Follow-up**	***p* ***
**anti-EBNA-1 IgG**	62 (96.9)	62 (95.3)	1.000	172.5 (0–200)	173 (0–200)	0.943
**anti-VCA IgG**	63 (98.4)	63 (98.4)	1.000	194.5 (0–200)	189 (0–200)	0.624
**anti-VCA IgM**	12 (18.8)	3 (4.7)	0.022	0 (0–4.45)	0 (0–2.55)	0.026
**anti-EA(D) IgG**	14 (21.9)	16 (25.0)	0.804	0 (0–200)	0 (0–200)	0.527
**anti-EA(D) IgM**	14 (21.9)	5 (7.8)	0.022	0 (0–2.83)	0 (0–3.51)	0.006
	**RA-B**					
**anti-EBV Abs**	**Seroprevalence**			**Titer**		
**Time**	**Baseline**	**6 Month Follow-up**	***p* ***	**Baseline**	**6 Month Follow-up**	***p* ***
**anti-EBNA-1 IgG**	33 (91.7)	34 (94.4)	1.000	124 (0–200)	127.5 (0–200)	0.112
**anti-VCA IgG**	36 (100.0)	36 (100.0)	NA	160 (18–200)	189 (90–200)	0.006
**anti-VCA IgM**	5 (13.9)	1 (2.8)	0.219	0 (0–5.57)	0 (0–1.48)	0.075
**anti-EA(D) IgG**	7 (19.4)	15 (41.7)	0.021	0 (0–175)	0 (0–195)	0.006
**anti-EA(D) IgM**	3 (8.3)	2 (5.6)	1.000	0 (0–7.30)	0 (0–1.93)	0.500

***** for the level of significance of 0.05 according to Chi-square test for seroprevalence and Mann-Whitney test for antibodies titers.

**Table 3 biomedicines-11-02375-t003:** Clinical and laboratory characteristics of all RA patients and RA-A and RA-B subgroups during the 6-Month follow-up.

Characteristic	All RA Patients		*p* *	RA-A Patients		*p* *	RA-B Patients		*p* *
	Baseline *n* = 133	6 Months after *n* = 100		Baseline *n* = 80	6 Months after *n* = 64		Baseline *n* = 53	6 Months after *n* = 36	
**Disease Activity**									
Total number of painful joints, med (min-max)	11 (0–33)	3 (0–20)	<0.001 £	10 (0–33)	2 (0–8)	<0.001 £	14 (0–31)	5 (0–20)	<0.001 £
Total number of swollen joints, med (min-max)	7 (0–32)	1 (0–15)	<0.001 £	6 (0–32)	2 (0–8)	<0.001 £	9 (0–30)	1 (0–15)	<0.001 £
Total number of painful and swollen joints, med (min-max)	19 (0–65)	4 (0–35)	<0.001 £	16 (0–65)	1 (0–10)	<0.001 £	24 (0–61)	6.5 (0–35)	<0.001 £
Tenosynovitis, *n* (%)	45 (33.8)	18 (17.6)	<0.001 ¥	26 (43.3)	10 (16.7)	0.002 ¥	19 (45.2)	8 (19.0)	<0.001 ¥
Fatigue, *n* (%)	77 (57.9)	43 (41.3)	<0.001 ¥	46 (73.0)	26 (41.3)	0.001 ¥	31 (75.6)	17 (41.5)	<0.001 ¥
Morning stiffness, *n* (%)	99 (74.4)	66 (63.5)	<0.001 ¥	60 (95.2)	43 (68.3)	<0.001 ¥	39 (95.1)	23 (56.1)	<0.001 ¥
Morning stiffness duration (min), med (min-max)	60 (5–480)	30 (5–240)	<0.001	60 (5–320)	30 (5–240)	<0.001 £	120 (15–480)	30 (10–240)	0.001 £
PtGA = VAS patient, med (min-max)	60 (10–100)	20 (0–70)	<0.001 £	60 (10–100)	20 (0–70)	<0.001 £	70 (30–100)	30 (0–70)	<0.001 £
PrGA = VAS physician, med (min-max)	50 (10–90)	10 (0–60)	<0.001 £	50 (10–90)	10 (0–50)	<0.001 £	50 (10–90)	10 (0–60)	<0.001 £
VAS pain, med (min-max)	60 (10–100)	10 (0–70)	<0.001 £	60 (10–100)	10 (0–60)	<0.001 £	60 (30–90)	10 (0–70)	<0.001 £
DAS8-ESR, med (min-max)	5.46 (1.53–8.04)	3.19 (0.90–7.70)	<0.001 £	5.12 (1.53–8.04)	3.28 (0.90–5.76)	<0.001 £	5.63 (5.10–6.63)	2.99 (1.19–7.70)	<0.001 £
DAS28-CRP, med (min-max)	4.89 (2.01–7.80)	2.70 (0.10–8.00)	<0.001 £	4.59 (2.01–7.80)	2.77 (0.10–5.53)	<0.001 £	5.09 (2.13–6.46)	2.50 (0.30–8.00)	<0.001 £
SDAI, med (min-max)	16.0 (0.0–63.0)	4.5 (0.0–31.6)	<0.001 £	12.30 (0–63)	4.5 (0–31.6)	<0.001 £	22.80 (6–62)	4.44 (0.00–27.10)	<0.001 £
CDAI, med (min-max)	18 (1–40)	4 (0–30)	<0.001 £	12.00 (1.0–26.0)	3 (0–30)	<0.001 £	22 (7–40)	6 (0–26)	<0.001 £
**Inflammatory markers**									
NLR, med (min-max)	2.57 (0.55–10.00)	2.40 (0.09–7.17)	0.002 £	2.61 (0.68–6.46)	2.47 (1.00–6.98)	<0.001 £	2.38 (0.55–10.00)	2.09 (0.09–7.17)	0.002 £
ESR (mm/h), med (min-max)	35.0 (6.0–120.0)	15.5 (2.0–80.0)	<0.001 £	37.0 (6.0–100.0)	15.0 (2.0–75.0)	<0.001 £	34.0 (10.0–120.0)	16.0 (3.0–80.0)	<0.001 £
CRP (mg/L), med (min-max)	15.0 (0.0–152.4)	3.25 (0.0–631.0)	<0.001 £	12.80 (0.0–84.1)	3.0 (0.1–63.1)	<0.001 £	19.3 (1.7–152.4)	4.0 (0.0–36.9)	<0.001 £
**Immunology parameters**									
ANA Abs titer, med (min-max)	40.0 (0.0–640.0)	/	NA	20.0 (0.0–640.0)	/	NA	40 (0–640)	/	NA
aCL IgG Abs positivity, *n* (%)	43 (32.3)	/	NA	23 (28.8)	/	NA	20 (37.7)	/	NA
aCL IgM Abs positivity, *n* (%)	41 (30.8)	/	NA	23 (28.8)	/	NA	18 (34.0)	/	NA
RF positivity (>20), *n* (%)	124 (93.2)	/	NA	74 (92.5)	/	NA	50 (94.3)	/	NA
ACPA Abs titer, med (min-max)	320.0 (0.0–500.0)	/	NA	284.5 (3.5–500.0)	/	NA	350 (0–500)	/	NA
**Quality of life**									
RAID, med (min-max)	5 (1–18)	3 (0–11)	<0.001 £	5 (1–16)	3 (0–11)	<0.001 £	5 (1–18)	3 (0–7)	<0.001 £
RAQoL, med (min-max)	12 (1–28)	6 (0–28)	<0.001 £	10.25 (1.0–28.0)	5.0 (0.0–27.0)	<0.001 £	13 (1–27)	7 (0–28)	0.010 £
HAQ, med (min-max)	1.125 (0.125–2.625)	0.650 (0.0–2.650)	<0.001 £	0.937 (0.125–2.125)	0.500 (0.000–2.375)	<0.001 £	1.250 (0.280–2.625)	0.750 (0.000–2.650)	<0.001 £

* for the level of significance of 0.05 according to McNemar’s test ¥ and Wilcoxon signed rank test £. Abbreviations: PtGA—Patient Global Assessment of Disease Activity; PrGA—Provider Global Assessment of Disease Activity; VAS—Visual Analog Scale; DAS28—Disease Activity Score With 28-Joint Counts; ESR—Erythrocyte Sedimentation Rate; CRP—C-reactive Protein; SDAI—Simplified Disease Activity Index, CDAI—Clinical Disease Activity Index; RAID—Rheumatoid Arthritis Impact of Disease index; RAQoL—Rheumatoid Arthritis Quality of Life Questionnaire; HAQ—Healthcare Access and Quality index; NLR—Neutrophil/Lymphocyte Ratio; ANA—antinuclear antibodies; aCL—anticardiolipin antibodies; ACPA—anti-citrullinated protein autoantibodies; NA—Not Applicable.

## Data Availability

Data are available upon reasonable request. All data relevant to the study are available on reasonable and justified request. Please contact the corresponding author. They are not publicly available to ensure the strict confidentiality of the patients’ data.

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
