# Peer review of "Uncovering the Role of Epstein–Barr Virus Infection Markers for Remission in Rheumatoid Arthritis"

_biomedicines, 2023, doi:10.3390/biomedicines11092375_

Round 1

Reviewer 1 Report

EBV infection has long been associated with the pathogenesis of RA, as introduced by the authors. Although this cohort study involves (a) the status of EBV infection and its’ changes during the six-month follow-up period of RA patients and (b) to explore whether the status of EBV infection and its’ changes is influenced by the different therapy approach and c) to evaluate whether some of the markers of EBV infection could be considered as markers of RA development and/or remission, novelty is my concern here.

I suggest this manuscript is more suitable to be published in MDPI's other journal" Diagnostics". 

The Quality of English is ok.

Author Response

Due to novelty brought by this paper and topic of the special issue „Recent Insight in Auto-Inflammatory and Autoimmune Diseases“, we believe that the manuscript would make a valuable contribution to the journal Biomedicines, and in particular mentioned special issue.

As described in the manuscript, previous studies that monitored EBV activity during the treatment of RA, used only the molecular markers of infection, including viral load:

  • Balandraud, N.; Guis, S.; Meynard, J.B.; Auger, I.; Roudier, J.; Roudier, C. Long-Term Treatment with Methotrexate or Tumor Necrosis Factor α Inhibitors Does Not Increase Epstein-Barr Virus Load in Patients with Rheumatoid Arthritis. Arthritis Care Res. 2007, 57, 762–767, doi:10.1002/art.22783.
  • Fujieda, M.; Tsuruga, K.; Sato, T.; Kikuchi, H.; Tamaki, W.; Ishihara, M.; Yamamoto, M.; Oishi, T.; Tanaka, H.; Daibata, M. Monitoring of Epstein–Barr Virus Load and Killer T Cells in Patients with Juvenile Idiopathic Arthritis Treated with Methotrexate or Tocilizumab. Rheumatol. 2017, 27, 66–71.
  • Balandraud, N.; Texier, G.; Massy, E.; Muis-Pistor, O.; Martin, M.; Auger, I.; Guzian, M.C.; Guis, S.; Pham, T.; Roudier, J. Long Term Treatment with Abatacept or Tocilizumab Does Not Increase Epstein-Barr Virus Load in Patients with Rheumatoid Arthritis - a Three Years Retrospective Study. PLoS One 2017, 12, e0171623.
  • Miceli-Richard, C.; Gestermann, N.; Amiel, C.; Sellam, J.; Ittah, M.; Pavy, S.; Urrutia, A.; Girauld, I.; Carcelain, G.; Venet, A.; et al. Effect of Methotrexate and Anti-TNF on Epstein-Barr Virus T-Cell Response and Viral Load in Patients with Rheumatoid Arthritis or Spondylarthropathies. Arthritis Res. Ther. 2009, 11, doi:10.1186/ar2708.
  • Valleala, H.; Kauppi, M.J.; Kouri, V.P.; Korpela, M. Epstein-Barr Virus in Peripheral Blood Is Associated with Response to Rituximab Therapy in Rheumatoid Arthritis Patients. Rheumatol. 2015, 34, 1485–1488, doi:10.1007/s10067-015-2992-0.

Observation of changes in the seroprevalence or level of anti-EBV antibodies was extremely rare. Thus, the novelty of our study is represented by evaluation of the dynamics of seropositivity and changes in anti-EBV antibody titer during defined therapeutic follow-up of RA patients. Thus, we demonstrated the significant decrease in serological markers of active EBV infection during the treatment period and in particular the identification of a lower titer of anti-EBNA1 IgG as a marker of entering into RA remission for the first time.

In addition, a significant reduction in seroprevalence and titer levels of anti-VCA IgM and anti-EA(D) IgM after 6 months from RA diagnosis suggested an important role of the immune response to viral infection at the beginning of the disease, while a significant decrease of anti-EA(D) IgG titers in the RA B population from our study suggested that inadequate response to methotrexate with disease activity indeks-DAS28>5.1 is followed by failed control of reactivation and/or persistent EBV infection.

Reviewer 2 Report

Estimated Authors,

Thank you for having provided this interesting article about the occurrence of EBV infection and its course in patients affected by R.A.

This pilot report, on a total of 133 patients, distributed between newly diagnosed cases and patients in regular follow up, with inadequate response to the first-line therapy, has found that in newly diagnosed patients some significant changes in seroprevalence pattern of EBV markers could be found, with the reduction of IgM and a steady representation of IgG. Therapy did not affect these features.

From my point of view, albeit of certain interest, the present paper is to date affected by several shortcomings that require a preventive assessment before the eventual acceptance.

First of all, the very design of the study is made unclear by the flow of data. 

According to the aims sub-section of the introduction, Authors designed the present study in order "a) to investigate the status of EBV infection and its’ changes during the six-month follow-up period of RA patients; b) to explore whether the status of EBV infection and its’ changes is influenced by the different therapy approach and c) to evaluate whether some of the markers of EBV infection could be considered as markers of RA development and/or remission. " The results are not reported consistently with the priority order pointed at by study Authors, and should be conveniently fixed.

Moreover, point a) and b) are improperly developed: Table 1 and corresponding data reporting have a clinical reporting that mostly focuses on RA. Also lab tests are usually associated with RA rather than with EBV infection. Data on EBV infection are reported by Table 3, and the percent values does not point to the potential differences between RA groups A and B, if any. This is a substantial flaw to the data reporting and should be fixed.

Regarding point c), the appropriate understanding of the results is affected by the way Authors reported their results in the main text. In order to achieve an appropriate appraisal, values for antibodies should be filled within table 1. 

Collectively, the shortcomings affecting points a to c impairs the reader to properly grasp whether some of the markers of EBV infection could be considered as markers of RA development and/or remission.

Some further and minor shortcomings:

- the present paper reports on a total of 133 cases; why 133? are we dealing with a series of consecutive patients (i.e. did the paper report on the whole experience of this centre on RA?) are we dealing with a specific subset of RA patients that guaranteed their participation and data sharing? Please provide a flow chart of included patients starting from the whole of patients usually followed by your centre during the study period.

- please remove notation marks for p values within Table 1 to 3. The reader could ascertain by him/herself whether a p value is significant or not, taking also into account that an unsolved debate involves the very same meaning of "significance" in biomedical studies.

- across the paper an inconsistent use of decimal notation is used (e.g. 0,001 vs. 0.001). Please make the paper consistent with the usual English notation (dots for decimals, comma for thousands).

The English text is globally appropriate. The text only needs some refinement on spelling and typos.

Author Response

  1. According to suggestion, we adapted the aims. Three of them were transformed into two more clearer statements.

It is now stated: “To fulfill the understanding of EBV infection dynamics during the course of RA, the goals of this research were: a) to investigate the status of EBV infection and whether its’ changes during the six-month follow-up period are influenced by the different therapy approaches of RA patients; b) to explore whether the status of EBV infection and its’ changes is influenced by the different therapy approach and c) to evaluate whether some of the markers of EBV infection could be considered as markers of RA development and/or remission.”

  1. Based on the reviewer’s suggestion, we reordered the tables in order to be in accordance with the aims of the study.
  2. We added seroprevalence for RA A and RA B subgroups as well as the titer values for RA, RA A, and RA B in Table 2.
  3. The interpretation of results for RA A and RA B subgroups were stated as follows within the 3.2 subsection:

- Analyzing the two groups of patients separately (RA A and RA B), seroprevalence of anti-VCA IgM and anti-EA(D) IgM showed a significant reduction in RA A patients (p=0.022 and p=0.022, respectively), while a significant decrease of anti-EA(D) IgG seroprevalence was detected in the RA B subgroup (p=0.021) during the 6 month follow-up.

- RA A patients showed significant decrease in anti-VCA IgM, anti-EA(D) IgM (p=0.026 and p=0.006, respectively), while in RA B patients the titers of anti-VCA IgG and anti-EA(D) IgG decreased significantly during the 6 month follow-up (p=0.006 and p=0.006, respectively).

- There was no difference in EBV EBNA1 DNA presence during the 6 month follow up in RA A and RA B subgroups as well (p=0.087 and p=0.375).

- There was no difference in the status of EBV infection in neither in RA A nor in RA B subgroups (p=1.000 and p=0.087, respectively).

  1. We completely reconstructed the 3.4 sub-section of results in order to make it easier to read and understand and eliminated redundant information.

Finally, we performed a logistic regression analysis in order to analyze factors associated with RA remission. RA remission was defined according to DAS28-CRP as the primary endpoint. Also, RA remission was defined as the presence of at least one of the following four criteria: DAS28-ESR, DAS28-CRP, SDAI, and CDAI as secondary endpoint. In all RA patients, anti-EBNA1 IgG antibodies titer level at baseline could be considered as a potential marker regardless of age and gender (OR=0.998, 95%CI OR=0.98-0.99, p=0.030). Also, the same anti-EBV antibody titer level could be considered as a marker of secondary endpoint regardless of age and gender (OR=0.99, 95%CI OR=0.98-0.99, p=0.038).This parameter was not a significant predictor of re-mission in newly diagnosed RA patients, but it could be considered as a significant marker of the primary endpoint, as well as, secondary endpoint in RA patients on biological therapy regardless of age and gender (OR=0.986, 95%CI OR=0.97-0.99, p=0,016 and OR=0.988, 95%CI OR=0.98-0.99, p=0,041).

Minor shortcomings:

- the present paper reports on a total of 133 cases; why 133? are we dealing with a series of consecutive patients (i.e. did the paper report on the whole experience of this centre on RA?) are we dealing with a specific subset of RA patients that guaranteed their participation and data sharing?

This study included succesively all RA patients who had met predefined inclusion criteria during the study period. That period was, unfortunatelly, during COVID-19 pandemic and the Institute of Rheumatology, Belgrade, turned into COVID-19 hospital in several episodes between June 2020 and June 2022.  It means that our hospital was unable to cope RA patients at all. At last, we achieved to collect all patients that had come to our hospital between COVID-19 waves in Serbia.

Please provide a flow chart of included patients starting from the whole of patients usually followed by your centre during the study period.

Together with the explanation on the previous question, it might be incorrect or even unrelevant to estimate the whole of patients followed by our centre during this period. Protocols for selecting hospitals and doctors for the treatment of non-COVID patients were frequently and dramatically changed, as we previusly described.

- please remove notation marks for p values within Table 1 to 3. The reader could ascertain by him/herself whether a p value is significant or not, taking also into account that an unsolved debate involves the very same meaning of "significance" in biomedical studies.

We removed the notation (bold) for p values within Table 1-3

- across the paper an inconsistent use of decimal notation is used (e.g. 0,001 vs. 0.001). Please make the paper consistent with the usual English notation (dots for decimals, comma for thousands).

We corrected the issue. It is now homogeneous.

Reviewer 3 Report

The manuscript by Banko et al. is focused on the assessment of EBV infection as a risk factor and evaluation of some mechanisms in RA patients. The manuscript contains some novel data, is well written and easy to follow. Nonetheless, I would like to encourage the authors to think about the title. The current one is very broad. In my opinion it should be more specific demonstrating the major result(s) of the study.

Author Response

According to the suggestion the title has been modified in:

„Uncovering the Role of Epstein-Barr Virus Infection Markers for Remission in Rheumatoid Arthritis“

Round 2

Reviewer 1 Report

As suggested previously, novelty is my concern regarding this manuscript. I think this paper is no problem to be published, however, "Diagnostics" is much more suitable for this paper. Of course, the decision depends on the Editor. 

Author Response

Based on previous description about the novelty and invitation from the editors of Biomedicines, we consider that the manuscript could contribute to this journal and special issue Recent Insight in Auto-Inflammatory and Autoimmune Diseases. However, we agree with the reviewer that the decision is up to the editors.

Reviewer 2 Report

Estimated Authors,

thank you for the accurate adjustments you did perform according to my previous review. The paper has been substantially improved and, from my point of view, is only a small step away from being accepted for publication.

While substantial issues have been solved after the re-working of results section, some formal ones still remain when dealing with discussion section. Not as a whole, by the way, but only regarding section from row 321 to row 368. Don't blame for the following comment the present reviewer, but I think that the aforementioned section is quite difficult to follow, and would deserve a substantial reworking in the flow of the main text. After that improvement, the paper could be accepted.

See above: a quite large section of the main text (row 321 to row 368) would require some adjustment before the eventual acceptance.

Author Response

The section from row 321 to row 368 has been improved. The sentences were shortened, their order was corrected and a small number of spelling mistakes were corrected also. We believe that in this way the text is simplified for reading and understanding.